# Maresin1 Ameliorates Sepsis-Induced Microglial Neuritis Induced through Blocking TLR4-NF-κ B-NLRP3 Signaling Pathway

**DOI:** 10.3390/jpm13030534

**Published:** 2023-03-16

**Authors:** Huiping Wu, Ying Wang, Haiyan Fu, Lili Ji, Na Li, Dan Zhang, Longxiang Su, Zhansheng Hu

**Affiliations:** 1School of Medicine, Soochow University, Suzhou 215006, China; 2Intensive Care Unit, The First Affiliated Hospital of Jinzhou Medical University, Jinzhou 121001, China; 3Operating Room, The First Affiliated Hospital of Jinzhou Medical University, Jinzhou 121012, China; 4Emergency Department, The First Affiliated Hospital of Jinzhou Medical University, Jinzhou 121012, China; 5Department of Critical Care Medicine, Peking Union Medical College Hospital, Beijing 100730, China

**Keywords:** sepsis, microglia, Maresin1 (MaR1), TLR4-NF-κB, NLRP3 inflammasome

## Abstract

**Objective**: Neuroinflammation is a major etiology of cognitive dysfunction due to sepsis. Maresin1 (MaR1), identified as a docosahexaenoic acid (DHA)-derived metabolite from macrophages, has been demonstrated to exhibit potent neuroprotective and anti-inflammatory effects. Nevertheless, detailed functions and molecular mechanism of MaR1 in sepsis-induced cognitive dysfunction has not been fully elucidated. Here, we aimed to investigate potential neuroprotective effects of MaR1 on microglia-induced neuroinflammation in sepsis-induced cognitive impairment and to explore its anti-inflammatory mechanism. **Methods:** Different doses of MaR1 were administered to septic rats by via tail vein injection. The optimal dose was determined based on the 7-day survival rate of rats from each group. derived from macrophages with both anti-inflammatory to observe the ameliorative effects of MaR1 at optimal doses on cognitive dysfunction in septic rats. The effects of MaR1 on neuroinflammation-mediated microglial activation, neuronal apoptosis, and pro-inflammatory cytokine productions were in vivo and in vitro assayed, using Western blot, ELISA, TUNEL staining, Nissl staining, and the immunofluorescence method. To further elucidate anti-inflammatory machinery of MaR1, protein expressions of NLRP3 inflammatory vesicles and TLR4-NF-κB pathway-related proteins were subjected to Western blot assay. **Results:** After tail vein injection of MaR1 with different doses (2 ng/g, 4 ng/g, 8 ng/g), the results showed that 4 ng/g MaR1 treatment significantly increased the rats’ 7-day survival rate compared to the CLP controls. Therefore, subsequent experiments set 4 ng/g MaR1 as the optimal dose. Morris water maze experiments confirmed that MaR1 significantly reduced space memory dysfunction in rats. In addition, in CLP rats and LPS-stimulated BV2 microglia, MaR1 significantly reduced activated microglia and pro-inflammatory cytokines levels and neuronal apoptosis. Mechanically, MaR1 inhibits microglia-induced neuroinflammation through suppressing activations of NLRP3 inflammatory vesicles and TLR4-NF-κB signal pathway. **Conclusion:** Collectively, our findings suggested that MaR1 might be a prospective neuroprotective compound for prevention and treatment in the sepsis process.

## 1. Introduction

Sepsis has been identified as a life-threatening organic dysfunction in which the host immune response system is disturbed due to infection [1]. The pathogenesis of sepsis is extensive and complex, mainly including pathophysiological processes such as imbalance of inflammatory response, mitochondrial impairment, immune dysfunction, coagulation disorders, neuroendocrine changes, endoplasmic reticulum stress (ERS), and autophagy, leading to ultimate organ diseases [2,3,4]. Sepsis is a systemic inflammatory disease in which multiple organs are affected simultaneously or sequentially [5]. Among the organs involved, the brain is the most susceptible organ, manifesting sepsis-induced brain dysfunction, which is closely associated with mortality and cognitive impairment [6]. Microglia are resident macrophages in the parenchyma of the CNS (central nervous system) and perform essential roles in neuroinflammatory regulation and host immune defense responses [5,7]. They are the first physiological barrier and defense of the brain to protect the CNS from damage [8]. During the development of sepsis, peripheral inflammatory cytokines enter the brain through the disrupted blood–brain barrier (BBB), causing microglia to activate and secrete bulks of inflammatory factors (such as IL-6, IL-1β, and TNF-α), leading to septic encephalopathy (sepsis-associated encephalopathy, SAE) [9,10]. NF-κB activates the transcription and expression of multiple genes in the cell nucleus, for example, nod-like receptor protein 3 (NLRP3) in microglia. NLRP3 can continuously activate and amplify the inflammatory response, and the enhancement of inflammatory response further exacerbates neuronal cell damage, ultimately leading to individual cognitive dysfunction [11,12].

The regression of inflammation is the result of the body’s self-regulation, in which endogenous pro-inflammatory mediators (SPMs) play a key role. The N-3 polyunsaturated fatty acids (PUFAs) belong to long-chain unsaturated fatty acids, mainly composed by DHA and EPA [13,14]. Maresin1 (MaR1) is a newly identified metabolite of 22-carbosahexaenoic acid (DHA) derived from macrophages with both anti-inflammatory and pro-inflammatory regressive effects [15,16]. Studies have revealed that MaR1 can inhibit the production of pro-inflammatory factors by suppressing neutrophil infiltration, enhancing macrophage phagocytosis and simultaneously blocking NF-κB pathway activation, exerting protective effects in a variety of chronic inflammatory diseases [17]. MaR1 was also found to inhibit acute lung-injury-induced neutrophil infiltration and adhesion in mice and downregulate the generation of inflammatory mediators [18]. Studies have demonstrated that MaR1 is effective in preventing atherosclerosis and could be used as a new potential therapeutic strategy to eliminate arterial inflammation [19]. Nevertheless, the role of MaR1 in septic rats and its regulation on microglia in septic rats remains to be explored.

## 2. Methods and Materials

### 2.1. Construction of Sepsis-Induced Encephalopathy Rat Model and Experimental Groupings

SD rats (male, adult, *n* = 80, weight ranging from 180–220 g) were obtained from the Animal Center of Suzhou University (Suzhou, China) and maintained at 22–25 °C and humidities of 40–50%. Water and food were readily available.

A total of 60 SD rats were randomly selected and anesthetized, followed by intraperitoneal injection of 10% sodium pentobarbital. Rats were fixed on the operating table with adhesive tape. Briefly, after disinfecting the abdominal skin of rats with povidone iodine solution, a 2 cm longitudinal incision was made along the midline of the rat’s abdomen, the abdominal cavity was opened, the cecum was found, the contents of the cecum toward the distal cecum were gently squeezed, and the cecum was ligated at the root of the cecum. Finally, we used an 18GG puncture needle to puncture the root of the cecum twice from the mesentery to the free side of the mesentery, squeezed a small amount of feces into the rat’s abdominal cavity, returned the cecum to the abdominal cavity, kept the incision free of feces, and sutured the abdominal wall layer by layer. Ringer’s solution (5 mL/100 g) was injected subcutaneously to prevent shock [20,21,22]. Rats in the sham-operated group were not subject to cecum ligation perforation (CLP).

Subsequently, 48 rats were randomly chosen and subdivided equally into 3 groups. At 24 h after surgery, different doses of MaR1 (2 ng/g, 4 ng/g, and 8 ng/g) were injected via the tail vein, respectively. All rats were allowed free access to food and water. All animal experiments involved in this study were conducted in accordance with the Guidelines for the Care and Use of Laboratory Animals of the National Institutes of Health (8th edition. Washington (DC): National Academies Press (US); 2011). The project was approved by the Animal Ethics Committee of Jinzhou Medical University (Jinzhou, China). Ethical number is 2021030501.

### 2.2. BV2 Microglia Cell Culture

Immortalized rat microglial cell line (BV2) was obtained from Shanghai Zhongqiao Xinzhou Biotechnology Co., Ltd. (Shanghai, China). Briefly, BV2 cells were cultured at 37 °C with 5% CO_2_ in high-glucose DMEM (Gibco, 11965126, USA, Grand Island) containing 5% FBS and (Fetal bovine serum) supplemented with100 U/mL streptomycin/penicillin (Gibco, 15140148, USA). All cells were divided into the Vehicle group, 10 mg/mL; the LPS group, 0.1 μmol/L; the MaR1+10 mg/mL LPS group; and the 0.5 μmol/L MaR1+10 mg/mL LPS group plate until the cellular density reached 80–90%.

### 2.3. 7-Day Survival Rate Analysis

After the above CLP rat model was established, MaR1 was injected via the tail vein, and the survival time of rats in each group was recorded and analyzed within 7 days.

### 2.4. MWM Test

The Morris water maze (MWM) test was performed on the 8th day after surgery to assess rat spatial learning and memory capabilities. As previously described by Lissner et al. [23], the test was carried out in a circular pool (approximately 120-cm in diameter and 50-cm height) with water kept around 25 °C. An invisible and fixed platform (approximately 10 cm in diameter) was submerged about 1 cm beneath the water surface. On Day 0, all animals swam freely in the pool without a platform for 1 min to familiarize themselves with the experimental environment and apparatus. On Days 1–5 of training, rats were subject to 4 trials/d, lasting 1 min for spatial acquisition and each with a 15-min intertrial interval, during which rats were gently released into the water from the edge of the pool. The experiment would be automatically terminated once a rat climbed onto the platform. A different starting position was used for each trial. If the rat could not reach the platform in 60 s, it would be gently guided back within the remaining time. During each experiment, both escape latency and path length adopted by rats for finding the hidden platform were recorded, and the animals’ performance was analyzed in a comprehensive manner. On Day 6, a probe trial was performed for memory consolidation evaluation, in which the platform was removed and animals were allowed to swim freely for 1 min. The number of times each rat traversed the original platform and the time spent on exploring were recorded.

### 2.5. Immunofluorescence Staining

Tissues were fixed with 4% paraformaldehyde by cardiac perfusion. Then, the tissue was placed in 4% paraformaldehyde and e a post-fixation was performed for 2 h at 4 °C. After gradient dehydration with 20% and 30% sucrose solution, the rats’ brain tissues were OCT-embedded, and coronal sections (10 μm) were prepared using a frozen slicer (CM1900Leica Microsystems, Wetzlar, Germany).

Slices were sealed with 5% BSA and 0.1% TritonX100 at RT for 1 h and incubated with Iba1 Rabbit pAb antibody (Servicebio,1:500, CHN, Wuhan) overnight at 4 °C. After washing, samples were incubated with the second antibody (FITC conjugated Goat Anti-Rabbit IgG, Servicebio, 1:500, USA) for 2 h. Then, 50 μL anti-fluorescence quenching sealing solution (S2110, including DAPI) was put on the tissue section and covered with cover glass. Images were taken using a microscope (Olympus, IX83, Japan) under the identical light intensity and exposure period.

### 2.6. Nissl Staining

For Nissl staining, sections (10 μm) were stained with a Nissl solution and subsequently dehydrated with gradients of alcohol (70%, 80%, 90%, and 100%). Finally, slides were washed with xylene and covered with coverslips. All images were taken by a microscope (Olympus, IX83, Japan) as described above. ImageJ (version is 1.8.0.112) was used to calculate the number of Nissl staining neurons.

### 2.7. TUNEL Staining

Slides were heated for another 10 min at RT and circled with a set of paintbrushes then fixed with 4% paraformaldehyde. After washing, samples were incubated in permeable solution (0.1% citric acid + 0.1% Triton X-100) at RT for 10 min. The TUNEL staining solution (Enzyme:label = 1:9) was prepared in advance, and the slides were placed in a light-proof box. Then, 50 μL TUNEL staining solution was added and incubated at 37 °C for 1 h to avoid light, followed by washing 3 times. Finally, slides were sealed with DAPI-containing tablets then dried away from light, and images were taken under an inverted fluorescence microscope (Olympus, IX83, Japan) and subjected to ImageJ (version is 1.8.0.112) software analysis.

### 2.8. Western Blot

To detect protein levels of NLRP3 and inflammatory and apoptosis factors in the hippocampus, fresh hippocampal tissues were sampled and homogenized in lysis buffer containing protease inhibitors (Roche, Basel, Switzerland), followed by centrifugation at 4 °C, 12,000× *g* for 20 min. Seahorse homogenate supernatant was collected. Protein concentrations were assayed with a BCA Protein Assay Kit (Beyotime, Shanghai, China). Samples were loaded to SDS–PAGE gels (10%, 30 μg/lane) after boiling in Loading buffer (Solarbio, Beijing, China) at 100 °C for 5 min, then subjected to electrophoresis at 80 V for 50 min followed by 120 V for 30 min. Proteins were transferred to a PVDF membrane (Millipore, USA, Massachusetts) under 250 mA condition for 90 min. The membrane was blocked with TBST buffer supplemented with 5% skim milk powder at RT for 2 h and subsequently incubated with specific primary antibodies at 4 °C overnight. Antibodies used in this study included anti-NLRP3 polyclonal antibodies (rabbit, 1: 3000, Invitrogen, USA), IL-18 (rabbit, 1:2000, Invitrogen, USA), IL-1β (rabbit, 1:1000, CST, MA, USA), cleaved caspase1 (rabbit, 1:1000, CST, USA), TLR4 (rabbit, 1:2000, Sigma, Germany), p-NF-KB p65 (rabbit, 1:1000, CST, USA), NF-κB p65 (rabbit, 1:1000, CST, USA), cleaved caspase3 (rabbit, 1:500, Abcam, UK), Iba-1 (rabbit, 1:1000, Abcam, USA), and β-actin (rabbit, 1:5000, CST, USA). After 16 h incubation, the membranes were equilibrated at RT for 30 min. After washing, samples were incubated with second antibody (HRP-conjugated Affinipure Goat Anti-Rabbit IgG, 1:5000, Proteintech, CHN) for 2 h at RT. Finally, proteins were visualized using a Western Bright ECL Spray (Solarbio, Beijing, China), and the intensity was analyzed by ImageJ (version is 1.8.0.112).

We use β-action as the standard, and the gray value of all stripes is subtracted from the gray value of the standard. In the process of statistical analysis, we calibrated the gray value of all sham groups to 1 and then compared other groups with sham groups.

### 2.9. ELISA

After supernatant of brain tissue and cells were obtained, total protein concentrations were determined by a BCA Kit as described above. Total proteins were extracted and L-18, TNF-α and IL-1β were, respectively, analyzed using an ELISA Kit (Solarbio, Beijing, China), according to the manufacturer’s instructions. Samples were tested at least in duplicate.

### 2.10. Statistical Analysis

All statistical analysis was carried out with GraphPad software (Ver. 9.0). Data were expressed as the mean ± SD. The statistical differences were calculated by One-way ANOVA, followed by the Fisher least significant difference (LSD) *t*-test (student’s). A *p* < 0.05 value was set as the default to be statistically significant.

## 3. Results

### 3.1. MaR1 Treatment Increases the 7-Day Survival Rate of Septic Rats

The optimal treatment dose of MaR1 was determined separately before preparing the MaR1 reagent [24]. Figure 1 shows the effect of MaR1 on septic rats’ 7-day survival rate. Considering the maximum survivals and first-time mortality together, the mortalities of MaR1 treated with concentrations of 2 ng/g and 8 ng/g were significantly much lower than those detected from the CLP groups treated with other test concentrations. Treatment with 4 ng/g MaR1 was the most effective and significantly improved the survival rate compared to the CLP group. Rats in the sham-operated control group did not die during the 7-day experiment, whereas rats in the CLP-treated group died within the first 60 h. Accordingly, 4 ng/g of MaR1 was set as the optimal therapeutic dose in all subsequent experiments.

### 3.2. MaR1 Significantly Improves Learning and Memory Deficits in Septic Rats

To evaluate whether the 4 ng/g of MaR1 could improve the learning and memory deficits in septic rats, behavioral experiments were first carried out [25]. On Day 8 after the CLP or sham operation, we performed MWM on rats for 6 consecutive days to assess the animals’ spatial learning and memory. In navigation trials, no significant difference was observed among the Sham, CLP, Sham+4 ng/g MaR1, and CLP + 4 ng/g MaR1 groups on Day 1–2. In the following Day 3–5, the evasion latency and path length were much higher in the CLP group than those detected from the Sham group, while no significant difference was observed between the Sham + 4 ng/g MaR1 and the Sham groups (Figure 2A,B). Compared to the CLP group, escape latency and path length of the CLP + 4 ng/g MaR1 group were significantly much lower (Figure 2A). In addition, spatial exploration experiments were performed on Day 6 to assess the memory of the four groups. Results showed that rats in the CLP group spent significantly less time on swimming in the target quadrant, and swam mainly in the tank therefore failing to cross the original platform position (Figure 2C). Compared to the CLP group, rats in the CLP + 4 ng/g MaR1 group spent longer time in the target quadrant and traversed the platform more often (Figure 2C,E). In addition, compared to the Sham group, no significant difference was observed in dwell time and the number of platform crossings between the Sham + 4 ng/g MaR1 and the Sham + 4 ng/g MaR1 groups (Figure 2D,E). These results demonstrated that treatment of 4 ng/g MaR1 significantly improved cognitive dysfunction in CLP rats. Therefore, 4 ng/g MaR1 could alleviate the cognitive decline in experimental sepsis.

### 3.3. MaR1 Blocks the Expression of Pro-Inflammatory Factors by Inhibiting Microglia Activation in the Hippocampal Region of CLP Rats

A constant signature of septic encephalopathy was identified as microglia activations [26,27,28]. Here, the treatment effect of MaR1 (4 ng/g) on hippocampal microglia activation in the CLP rats was evaluated by immunofluorescence. As indicated in Figure 3A,B, the increase of the number of Iba-1-positive microglial cells was present in the hippocampal region of rats in the CLP group, while 4 ng/g of MaR1 treatment significantly reduced the number of positive microglia. Meanwhile, Iba-1 protein expression was significantly lower in 4 ng/g MaR1 group than that detected from the CLP group (Figure 3C,D).

We further investigated whether 4 ng/g MaR1 reduced levels of pro-inflammatory cytokines in the hippocampus of CLP rats. ELISA analysis results showed that IL-6, IL-1β, TNF-α, and IL-18 were significantly increased in hippocampal tissues of rats in the CLP group. However, compared to CLP rats, 4 ng/g MaR1 significantly inhibited levels of IL-6, IL-1β, TNF-α, and IL-18 (Figure 3E). Overall, these data suggested that 4 ng/g MaR1 administration alleviated microglial activation and neuroinflammation in the hippocampal region of CLP rats.

### 3.4. MaR1 Effectively Reversed Hippocampal Neuronal Apoptosis in Septic Rats

Sepsis-induced microglial activation combined with massive release of pro-inflammatory cytokines lead to hippocampal neuronal apoptosis and cleaved-caspase3 expression [29,30]. Western blot results showed that cleaved aspase3 were significantly reduced in the hippocampus of rats in the 4 ng/g MaR1 group compared to CLP rats (Figure 4A,B). Consistent with the above results, Nissl staining further demonstrated that 4 ng/g MaR1 significantly increased the number of normal hippocampal neurons in CLP rats (Figure 4C,D). TUNEL staining confirmed that 4 ng/g MaR1 significantly decreased the number of TUNEL-positive cells (Figure 4E,F), indicating 4 ng/g MaR1 treatment prevented hippocampal neuronal apoptosis in septic rats.

### 3.5. MaR1 Exerts Neuroprotective Effects through Blocking the Activation of the TLR4-NF-κB/NLRP3 Signaling Pathway

In the present study, we found that 4 ng/g MaR1 significantly reduced levels of IL-18, IL-6, TNF-α, and IL-1β. Furthermore, the maturation and secretion of inflammatory factors including IL-18 and IL-1β are mainly dependent on NLRP3 inflammatory vesicles. We therefore hypothesized that inhibition of NLRP3 inflammatory vesicle activation by 4 ng/g MaR1 might be a potential mechanism for suppressing pro-inflammatory cytokine release. We next evaluated the activation and expression of NLRP3 inflammatory vesicles by Western blot. Results showed that NLRP3 protein expression levels were significantly upregulated in the CLP group compared to those detected from the Sham group. In contrast, 4 ng/g MaR1 significantly decreased the NLRP3 expression. In addition, cleaved-caspase1, IL-1β and IL-18 was significantly upregulated in CLP rats compared to the Sham group. However, compared to the CLP group, 4 ng/g MaR1 significantly inhibited cleaved-caspase1, IL-1β, and IL-18 expressions in CLP rats (Figure 5A,B). These data suggested that MaR1 suppressed the activation of NLRP3 inflammatory vesicles in CLP rats.

Activation of NF-κB cell signaling is considered as an early event necessary for NLRP3 inflammatory vesicle activation [31,32,33]. Our data showed higher levels of p-NF-κB p65 in the CLP group compared to the Sham group, indicating that the NF-κB pathway was activated. We further examined the protein expression level of TLR4, since NF-κB was triggered by TLR4 activation. As expected, TLR4 expressions were significantly elevated in the CLP group. In contrast, 4 ng/g MaR1 administration treatment significantly downregulated TLR4 and NF-κB p65 phosphorylation (Figure 5C,D). Hence, our results suggested that TLR4/NF-κB cell signaling activation in CLP rats was largely inhibited by MaR1.

### 3.6. MaR1 Inhibits Microglia Activity and Morphological Changes in BV2 Cells by Suppressing LPS-Induced Pro-Inflammatory Mediators

Activated microglia produce large amounts of pro-inflammatory cytokines that induce neurotoxicity [34]. To further assess whether MaR1 suppress LPS-induced microglia activation, the number of Iba-1 positive cells was determined based on immunofluorescence analysis. Results showed that 10 mg/mL LPS induction stimulated microglia activation and increased the number of Iba-1 positive cells. Meanwhile, Iba-1 expression level was significantly decreased, and the best effect was achieved under 0.5 μmol/L MaR1 treatment (Figure 6A,B). Therefore, we used 0.5 μmol/L as MaR1 as the optimal treatment dose of MaR1 in all subsequent experiments. In addition, we observed under the microscope that normal microglia had small, rounded cytosol and few protrusions. In contrast, cells activated by 10 mg/mL LPS were larger, amoeboid in shape, and partially contracted in extension. In addition, 0.5 umol/L MaR1 treatment significantly reversed the abovementioned morphological changes of microglia, indicating that the pretreatment effect of MaR1 in sepsis was strongly associated with its inhibitory properties on microglial activation and morphological alterations.

### 3.7. MaR1 Attenuates LPS-Induced Activation of TLR4-NF-κB Cell Signalling in BV2 Cells

NLRP3 inflammatory vesicles play a major role during transcriptional regulation of pro-inflammatory cytokines [35]. To clarify the mechanism by which 0.5 μmol/L MaR1 regulates microglia activation, we first examined the effect of 0.5 μmol/L MaR1 on LPS-induced NLRP3 inflammatory vesicle activation in BV2 cells. Consistent with expectations, 0.5 μmol/L MaR1 significantly inhibited LPS-induced NLRP3, cleaved-caspase1, IL-1β, and IL-18 levels (Figure 7A,E). Collectively, these data suggested that anti-inflammatory effects of MaR1 were mediated through the inhibition of NLRP3 inflammatory vesicle activation in microglia.

The classical TLR4-NF-κB signaling pathway performs essential roles in modulating NLRP3 inflammatory vesicle activation [36]. Here, we investigated the potential effect of 0.5 μmol/L MaR1 on TLR4-NF-κB cell signaling in microglia. Western blot results indicated that 0.5 μmol/L MaR1 significantly inhibited the LPS-induced NF-κB p65 nuclear translocation (NT) process in BV2 microglia (Figure 7G). Simultaneously, 0.5 μmol/LMaR1 was found to significantly reduce the protein expression level of TLR4 in BV2 microglia stimulated by LPS (Figure 7A,F). In addition, by administering 2 μL TLR4 inhibitor TAK-242 in vitro, we found that the administration effect of 0.5 μmol/L MaR1 was comparable to that of the inhibitor-treated group (Figure 7). From this, we concluded that 0.5 μmol/L MaR1 exerts its anti-inflammatory effects mainly through inhibiting LPS-induced NLRP3 inflammatory vesicles activation in BV2 cells and blocking TLR4-NF-κB cell signaling.

## 4. Discussion

Here, we showed that MaR1 administration via tail vein injection significantly attenuated neuroinflammation in septic rats. In addition, MaR1 significantly inhibited the activity of BV2 microglia and LPS-induced inflammatory factor release in vitro. More importantly, anti-inflammatory effects of MaR1 were largely mediated by suppressing NLRP3 inflammatory vesicles in microglia. We also found that MaR1 exerts anti-inflammatory effects through inhibiting activation of NLRP3 inflammatory vesicles and blocking TLR4-NF-κB cell signaling in microglia.

During the development of sepsis, peripheral inflammatory cytokines enter the brain through the disrupted blood–brain barrier (BBB), causing microglia activation and secretion of multiple inflammatory factors, leading to central inflammatory response and septic encephalopathy (sepsis associated encephalopathy, SAE) [37,38]. Microglia have been identified as resident macrophages in CNS [39]. Microglia can cause multiple neurodegenerative diseases. Activated microglia release a range of pro-inflammatory factors [40]. Previous studies indicated that activated microglia can damage neurogenesis and are mainly associated with cognitive and mood dysfunctions [41]. Maresin1 (MaR1), a newly identified metabolite of 22-carbon hexaenoic acid (DHA) derived from macrophages, displays anti-inflammatory and pro-inflammatory effects [42]. Notably, MaR1 has been demonstrated to significantly inhibit macrophage infiltration and inflammation in the hippocampus. In the present study, MaR1 was found to significantly restore cognitive function in a septic-rat model by inhibiting microglia activation.

The main pathological characteristics of septic encephalopathy (SAE) are progressive memory loss (PML) and deterioration of cognitive function [12]. Since the hippocampus is one of the key regions responsible for memory, our study mainly focused on whether MaR1 in the hippocampus could reduce neuronal apoptosis induced by neuroinflammation [43]. In our study, CLP rats were injected via caudal vein with 2 ng/g, 4 ng/g, and 8 ng/g of MaR1 respectively, and the 7-day survival rates were calculated. Accordingly, the optimal dose for injection was determined to be 4 ng/g. Behavioral data further indicated that 4 ng/g MaR1 was sufficient to alleviate CLP rats’ spatial memory impairment and learning deficits. Consistent with previous studies, concerns regarding potential benefits of MaR1 in SAE treatment were also raised. Microglia activation plays an essential role in disease progression associated with neuroinflammatory processes. Activated microglia are mainly characterized by branching or enlarged cellular shapes yielding a broad-spectrum of pro-inflammatory cytokines (IL-1β, IL-18) and many other mediators, leading to ultimate neuronal damage in SAE [44]. Therefore, inhibition of microglia activation might be a key strategy. We further investigated whether MaR1 suppresses the activation of microglia and inflammatory factors. Immunological analysis demonstrated that the hippocampal region of CLP rats presented with substantial microglia activation, which could be significantly attenuated by MaR1 treatment [45]. Further studies indicated that MaR1 significantly inhibited the LPS-induced activation of BV2 microglia, confirming the early observations. At the same time, Western blot results demonstrated that IL-6, IL-18, TNF-α, and IL-1β were significantly elevated in activated microglia. These findings provide new evidence that MaR1 could be a promising drug for the treatment of neuroinflammation caused by microglia activation.

TLR4 is widely expressed on the surface of microglia in the CNS [46]. Studies suggest that TLR4 plays essential roles during the development and regulation of inflammation [47]. For example, TLR4 is involved in downstream molecular events such as NLRP3 and pro-IL-β gene transcription by promoting the activation of the NF-κB complex [48]. The NF-κB complex is a protein homo- or heterodimer composed of subunits p50 and p65 [49]. In normal conditions, p50 and p65 dimers of NF-κB are inactive in cytoplasm combined with IκB-family inhibitors [50]. The TLR4-triggered signaling transductions mediate IκB phosphorylation and degradation, enabling p65 subunit shuttling into the nucleus and bind to specific DNA shared sequences, ultimately leading to enhanced transcription of inflammation-associated proteins [51]. In this study, Western blotting was performed to assess whether MaR1 influences TLR4-NF-κB cell signal transduction both in vivo and in vitro. The results showed that TLR4 was activated in CLP rats and LPS-treated BV2 cells, which further promoted the activation of downstream NF-κB signal transduction. However, MaR1 significantly suppressed the upregulated expression of TLR4 and the subsequent NF-κB p65-mediated nuclear translocation, which is consistent with the in vitro administration of the TLR4 inhibitor TAK-242. We therefore proposed that MaR1 treatment on NLRP3 inflammatory vesicle activation is mediated through downregulation of TLR4-NF-κB signal transduction.

Neuroinflammation-induced neuronal apoptosis is considered to be the destiny of neurodegenerative neurons in sepsis which consequently exacerbates cognitive deficits [52]. Cleaved-caspase3 has been identified as an important factor contributing to neuronal apoptosis [53]. Here, we firstly examined effects of MaR1 on apoptosis-associated proteins in vivo. Results showed that cleaved-caspase3 was remarkably elevated in the hippocampus of CLP rats, while it was effectively inhibited after MaR1 intervention. Similarly, we obtained consistent findings that MaR1 effectively ameliorated hippocampal neuronal apoptosis in CLP rats as observed by Nissl staining. Overall, our data demonstrated that MaR1 attenuated neuroinflammation and reduced neuronal loss through blocking microglia activation, thereby rescuing cognitive impairment. These findings provide some new scientific evidence for promoting the practical application of MaR1 in the development of sepsis.

## Figures and Tables

**Figure 1 jpm-13-00534-f001:**
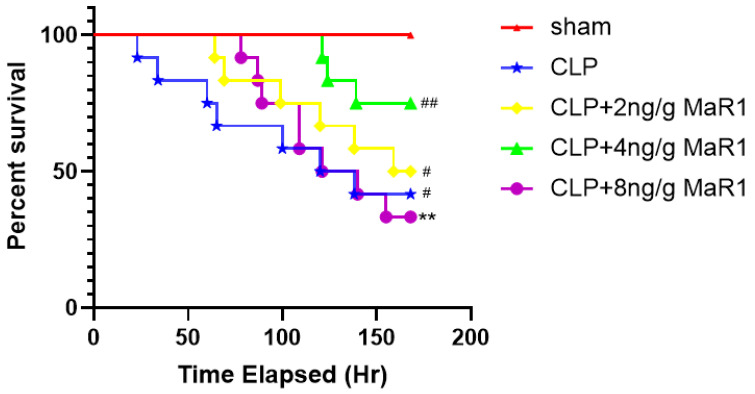
Effects of MaR1 treatment under different doses on mortality in septic rats. The median time CLP group is 129. The median time CLP+2 ng/g of the MaR1group is 163.5. The median time CLP+8 ng/g of the MaR1group is 130.4. The number of deaths in the sham group is 0. The number of deaths in the CLP group is 7. The number of deaths in the CLP+2ng/g of the MaR1 group is 6. The number of deaths in the CLP+4ng/g of the MaR1 group is 3. The number of deaths in the CLP+2ng/g of MaR1 group is 8; ** *p* < 0.01, compared to the Sham group; # *p* < 0.05, ## *p* < 0.01, compared to the CLP group (*n* = 12).

**Figure 2 jpm-13-00534-f002:**
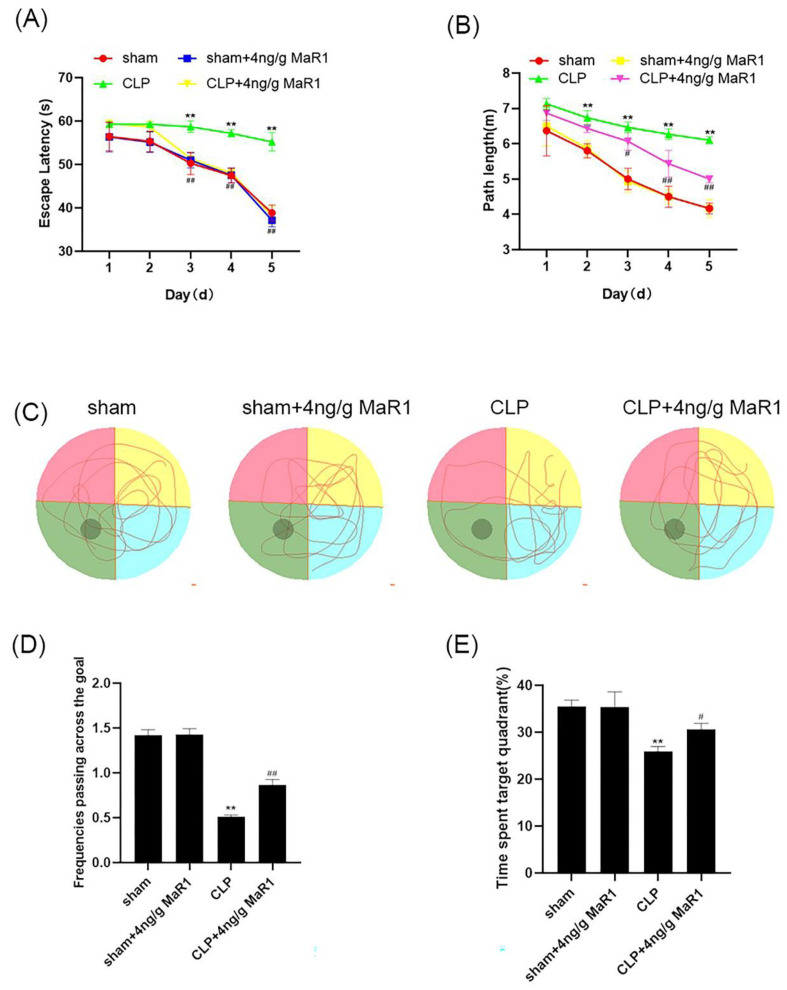
Four ng/g of MaR1 ameliorated learning and memory impairment in CLP rats: (**A**,**B**) evasion latency and path length of each group of rats were measured by a MWM test; 4 ng/g MaR1 shortened the distance and time for CLP rats to escape the incubation period; (**C**) representative swimming trajectory maps in space exploration experiments. Compared with CLP rats, rats spent more time in the third quadrant after 4 ng/g MaR1 treatment; (**D**,**E**) frequency of passage and time spent in the target quadrant during exploration trials. Data are expressed as mean ± SD (*n* = 10/group); ** *p* < 0.01, compared to the Sham group; # *p* < 0.05, ## *p* < 0.01, compared to the CLP group.

**Figure 3 jpm-13-00534-f003:**
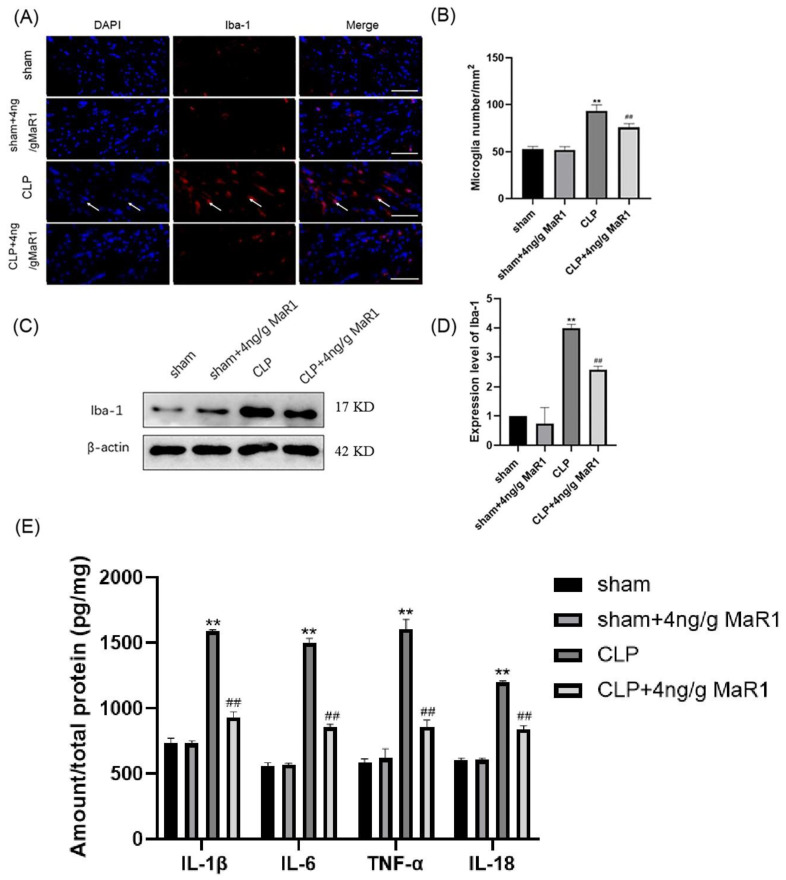
4 ng/g MaR1 reduced microglia activation and pro-inflammatory cytokines in CLP rats: (**A**,**B**) immunofluorescence images of Iba-1 (red)/DAPI (blue) co-localization in hippocampal region of CLP rats and the quantitative analysis of immunofluorescence positive cells (*n* = 5/group); after 4 ng/g MaR1 treatment, the number of Iba1 positive cells in the hippocampal CA1 region of CLP rats was significantly reduced; bar = 100 μm; (**C**,**D**) representative immunoblotting bands detected by anti-Iba-1 and β-actin antibodies; grayscale values of the immunoblot bands were quantified (*n* = 5/group); (**E**) levels of IL-6, IL-1β, TNF-α, and IL-18 in the hippocampus determined by ELISA (*n* = 4). Data are expressed as mean ± SD; ** *p* < 0.01, compared to the Sham group; ## *p* < 0.01, compared to the CLP group.

**Figure 4 jpm-13-00534-f004:**
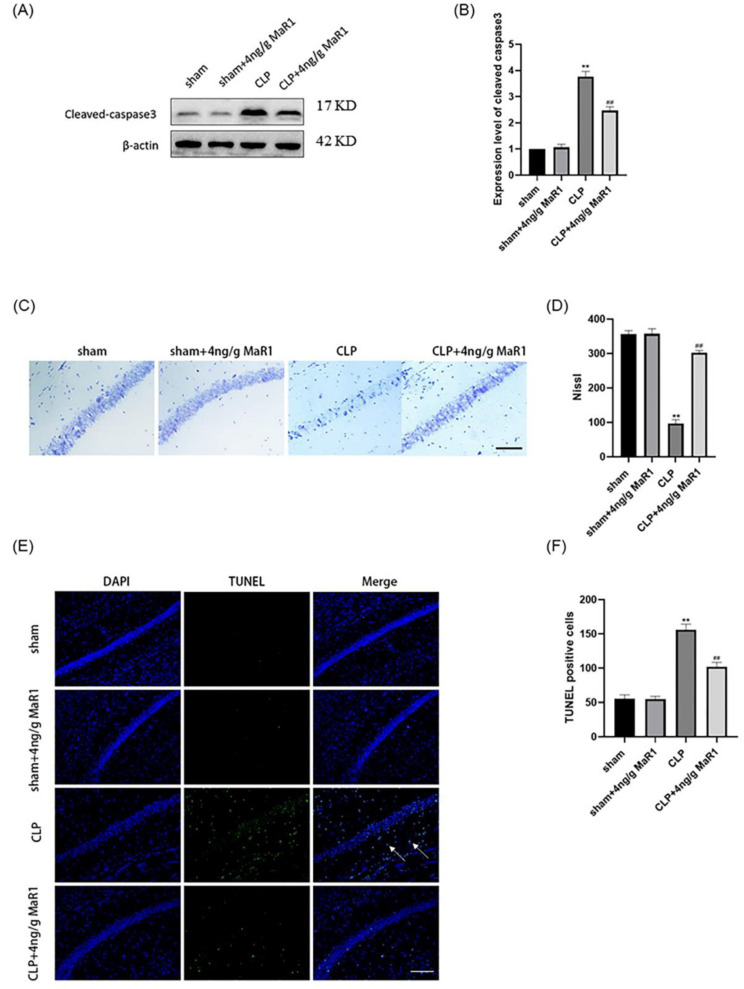
4 ng/g of MaR1 treatment prevents neuronal apoptosis in CLP rats: (**A**,**B**) protein expression levels of cleaved-caspase3 in the hippocampus of CLP rats; (**C**,**D**) nissl staining analysis of neuronal morphology and the number of normal neurons in the hippocampus of CLP rats, bar = 50 μm; (**E**,**F**) quantitative analysis of TUNEL-positive cells in the hippocampus of CLP rats by TUNEL staining, bar = 50 μm; ** *p* < 0.01, compared to the Sham group; ## *p* < 0.01, compared to the CLP group.

**Figure 5 jpm-13-00534-f005:**
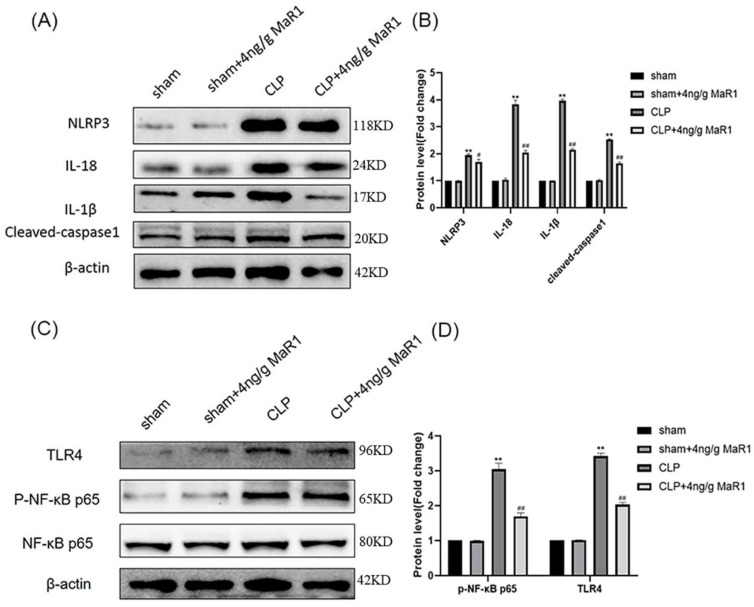
Activation of NLRP3 inflammatory vesicles and dominance of TLR4-NF-κB signaling by 4 ng/g MaR1 on CLP rat hippocampal homogenates: (**A**) medical use of NLRP3, IL-1β, IL-18, cleaved-caspase1, and β-actin; (**B**) quantitative standards into beta actin NLRP3, cleaved-caspase1, IL-1β, I-18; (**C**) use for TLR4, p-NF-κB p65, NF-κB p65, and β-actin antibody detection of representative immunoblotting; (**D**) quantitative levels of TLR4 and p-NF-κB p65 standardized by β-actin; data are expressed as mean ± SD (*n* = 4/group); ** *p* < 0.01, compared to the Sham group; # *p* < 0.05, ## *p* < 0.01, compared to the CLP group.

**Figure 6 jpm-13-00534-f006:**
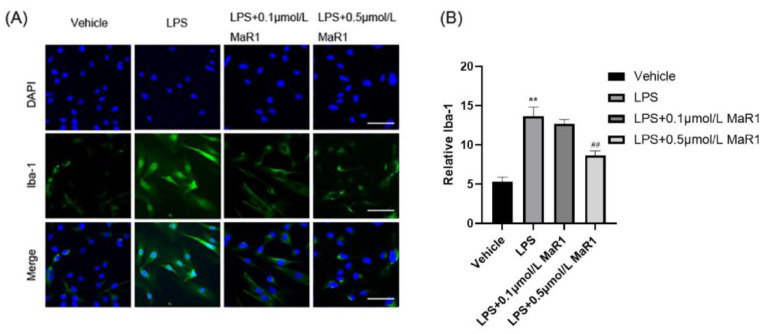
MaR1 treatment (0.5 μmol/L) inhibited LPS-induced microglia activation and morphological changes in BV2 cells: (**A**) LPS-induced fluorescent Iba-1 (green)/DAPI (blue) immunostaining, bar = 50 μm; (**B**) quantification of the Iba-1 immunofluorescence; all data are expressed as mean ± SD; ** *p* < 0.01, compared to the Vehicle group; ## *p* < 0.01, compared to the LPS group.

**Figure 7 jpm-13-00534-f007:**
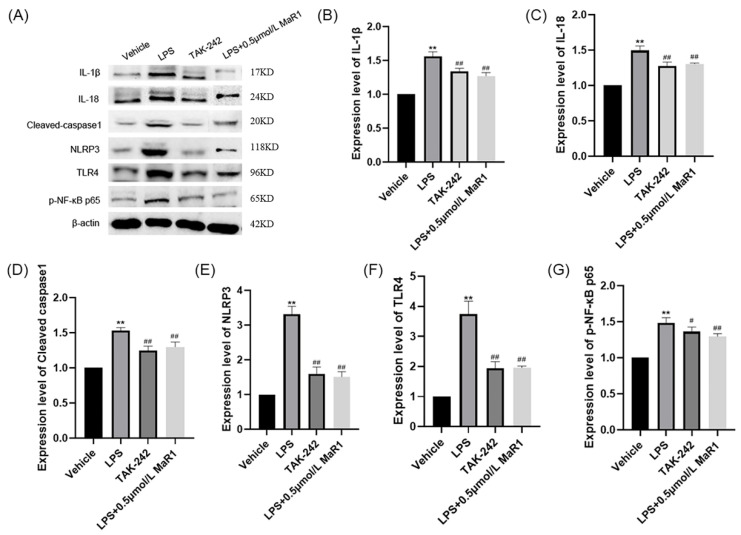
Levels of IL-1β, IL-18, cleaved-caspase1, TLR4, NLRP3, and pNF-κB p65 were determined by Western blot. (**A**) western blot images; (**B**–**G**)The β-actin was used as an internal reference, statistical bar graph of protein expression levels; ** *p* < 0.01, compared to the Vehicle group; # *p* < 0.05, ## *p* < 0.01, compared to the LPS group.

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
