# Peer review of "Maresin1 Ameliorates Sepsis-Induced Microglial Neuritis Induced through Blocking TLR4-NF-κ B-NLRP3 Signaling Pathway"

_jpm, 2023, doi:10.3390/jpm13030534_

Round 1
Reviewer 1 Report
The present manuscript focuses on determining the effects of maresin 1 on neuroinflammation in a rat model. The overall results show that this molecule could have an immunomodulatory protective effect in neuroinflammation induced by sepsis. Although, the objective of the study is well established, in my very personal point of view, the main problem of this study is the way the results are described and presented (figures), more importantly, the chronology of the results is not clear, for example, the first performed experiment was a survival rate where figure 1 shows that at 4ng/g of maresin, 50% of rats survived an average of 7 days and then, a Morris water maze was performed on day 8 after surgery. How many rats were used for that experiment? were rats expose to water and to swim after a very recent surgery that could infect the wound and alter the results? The ethical protocols for justifying such aggressive way to induce sepsis are not clear, besides, rats were not kept in sterile conditions after surgery, how the authors tested that sepsis was induced by the surgery and not for some other conditions?
Detailed comments follow.
Abstract
· The abstract is well structured and concise.
Introduction
· Introduction has a big problem with the order of the ideas and needs exhaustive English editing.
Methods
· In general, the methods section is well written and well-structured although it also needs English editing.
Results
· Results section is fairly structured; however, the description is a bit confusing, especially on the chronology of the results. More importantly, it is not well described which experiments were performed in brain tissue and which in cell lines.
Figures
· All figure legends lack clarity and need to be improved and explain with more detail the figure.
· Figure 1 lacks the n for each group.
· Figure 5 Why did authors not test p50 for NFKB?
· Figure 6 says that Iba-1 is stained in red and in the figure is observed in green.
Many details like that should be reviewed in all figures and legends.
Discussion
Discussion is well written; it seems like the manuscript was written by different persons, this section is better structured, however, it also needs a mild English revision.
There are some inconsistencies in this section, to mention few:
· iNOS is not a cytokine, it is an enzyme that mediates NO production.
· NF-KB is a transcription factor.
· Cognitive functions cannot be only determined by a Morris water maze and discussion should focus only on the cognitive functions (spatial memory) that this maze can detect.
Reviewer 2 Report
The manuscript entitles “Maresin1 ameliorates sepsis-induced microglial neuritis induced through blocking TLR4/NF-κ B/NLRP3 signaling pathway”, is novel work that allows us to observe how MaR1 attenuates neuroinflammation and reduces neuronal loss by blocking microglia activation, thus rescuing cognitive deterioration. Therefore, it provides evidence to consider the practical application of MaR1 in the development of sepsis. However, the manuscript requires a careful review of different formal aspects. Therefore, its publication with minor revisions is recommended.
1. Read the title carefully and modify it to make sense, for example, hyphenate TLR4/NF-κB/NLRP3.
2. consult the indications on References and modify them in the text, for example, replacing the references in the text with numbers. Review the rest of the instructions on how to write a manuscript (https://www.mdpi.com/journal/jpm/instructions#preparation “….In the text, reference numbers should be placed in square brackets [ ], and placed before the punctuation; …”
3. Carefully review all the text and indicate the meaning of the acronyms that appear in the text, for example FBS, etc.
4. Add the data from “Guidelines for the Care and Use of Laboratory Animals of the National Institutes of Health”, including ISSN, year of publication and endorsing body, etc.
5. Complete the methods performed. It is observed that some methods are very well described and others are not. For example, in section 2.5, what do you do after the incubation of the secondary? etc. In addition, they must indicate the reference number of the products used and the devices used to achieve the results, also indicating the commercial house and country. Please review the entire section
6. Indicate the references of the software used, for example those for capturing microscope images or image processing (ImageJ). Indicate the version used and the technical data. In the specific case of ImageJ, also indicate the Indicate also the tools used to count and the total number of elements counted in each case.
7. In section 2.5, after intracardiac fixation, you will do a post-fixation, right? Please correct the error and state the temperature at which you postfixed.
8. Remove the boldface from Figures in the text, and replace Fig. with Figure or Figueres
9. Figure 3. Provide Western Blot slides without treatment or clippings (you can put it as supplementary material). Put the larger letters in the 3B and 3D graphs
10. Figure 4. Provide Western Blot slides (you can put it as supplementary material) without treatment or clippings. Put the larger letters in graphs 4B, 4D and 4F.
11. Figure 5. Provide Western Blot slides (you can put it as supplementary material) without treatment or clippings. Put the larger letters in graphs 5B and 5D.
12. Figure 6. Put the larger letters in graph 6B. 6C photos are not good. Make new images that are better and increase the size of the photomicrographs,
13. Figure 7. Provide Western Blot slides without treatment or clippings (you can put it as supplementary material). Put the larger letters in the graphs
Reviewer 3 Report
The study by Wu and colleagues clearly and convincingly demonstrates that Maresin-1, a DHA metabolite shown to ameliorate skin inflammation through modulating the IL-23 pathway, potently suppresses signs of sepsis in an animal model and mechanistically modulates NLRP3 and TLR4/NF-κB signaling. Brain inflammation and neuronal death related to the systemic sepsis are similarly suppressed. A series of complementary approaches supports the authors mechanistic model by which Maresin-1 acts in vivo and in vitro. The study is well presented and appropriately illustrated.
The key concern is that the controls for treatments seem inadequate. The authors state, “Twenty-four hours after surgery, different doses of MaR1 (2 ng/g, 4ng/g, and 8ng/g) were injected via tail vein respectively.”
1. What is used as a carrier?
2. Control treatment at a minimum should include injection of the carrier solution alone, not simply untreated as it appears in almost all figures.
These issues need to be addressed in order to fully evaluate the results of this study.
Author Response
Please see the attachmen

Reviewer 4 Report
Authors have elaborated an interesting manuscript highlighting the anti-inflammatory potential of Maresin1 to inhibit microglia activation though canonical pathways. The manuscript is generally well written, though with many minor typos. The text attests knowledge in the field. However, I’ve found few inconsistencies in the results section, that can compromise the quality of the article. Therefore, I recommend the publication of this article after major revisions.
Please consider these minor & major questions:
Q1. Abstract: “Behavioral experiments were perfoemed”. Replace by “performed”
Q2. Introduction: “…derived from macrophageswith both anti-inflammatory”. Add a space between “macrophages” and “with”.
Q3. Methods: “Ringer, s solution”. Replace by “Ringer’s solution”.
Q4. Methods: “Rats in the sham-operated group were not subject to cecum ligation perforation..”
Please use the abbreviation (CLP) here, since it’s the first time you describe it in your manuscript. Also, remove the extra “.”
Q5. Methods: “2.2. cell and cell culture”
this section should be renamed, "BV2 microglia cell culture"
Q6. Methods: “were transplanyted into another 6-well”
Please, correct the typo.
Q7. Methods: “All rats were divided into Vehicle group, 10 mg/mL LPS group, 0.1 μmol/L MaR1+10mg/mL LPS group and 0.5 μmol/L MaR1+10 mg/mL LPS group.”
All rats?? This should be in the previous section, not here.
Please indicate the reference of the LPS in the methods. That may influence the interpretation of your results.
Q8. Methods: “(Millipore, , USA)”
Please, remove the extra comma.
Q9. Methods: “ANOVA, followed by Fisher least significant difference (LSD) t-test.”
Fisher LSD or t-test (student’s)? Please justify in the methods why you choose the appropriate test.
Q10. Results: Please add a brief possible explanation (can be in the discussion) for 8 ng/g being much less effective than 4 ng/g.
Q11. Results (Figure 3): (A) Which hippocampal region did you show? It was the same region for all conditions? This is important due to the microglial distribution through those regions, it may be different. (C) Add molecular weight of each protein indication (on side) of your western blot panels. Do this in every WB image in the manuscript. (E) Are you sure you don’t have proteins from other brain regions? It is not clear (even in the methods) how hippocampal tissue was separated and used.
Q12. Results (Figure 4): Provide an explanation for such improvement with nissl (discussion)??
Q13. Results (Figure 5): (B) Its strange that you have no changes in the WB image for Cleaved-caspase1 and the graph has such a significant decrease. Besides, the B-actin is much less dense in the lane of CLP+MaR1... This leads to my question:
Are the fold changes in the graph really obtained by normalizing to B-actin? From 4 different WBs? The SDs are surprisingly small to include the variations you show in the image.
Q14. Results (Figure 5): (D) Please describe exactly how you normalize p-NF-kB. To B-actin? or to NF-kB p65? or both?
Again, I see no changes in the image that could justify such a big difference observed in the graph. For TLR4, it makes more sense to me.
Q15. Results (3.6): “that 10 mg/ml LPS induction” That is a huge excessive LPS concentration!
You should check viability. And/or show from what reference you decided to use this concentration.
Q16. Results (3.6): “we all used 0.5 μmol/L as MaR1 as the optimal treatment dose of MaR1”
Can you explain how did you determine 0.5 umol/L (in vitro) from the 4ng/g (in vivo)?
Q17. Results (3.6): “cells activated by 10 mg/ml LPS were larger, amoeboid in shape, and partially contracted in extension.”
Images are too small and are not elucidative of this description.
Q18. Discussion: “In our study, CLP rats were injected via caudal vein with 2 ng/g and 4 ng/g of MaR1 respectively”
You forgot to mention 8ng/mL!
Besides, you should discuss why 8 has less effect than 4, as well as the possible side effects of MaR1 when inflammation is physiologically required.
Author Response
Please see the attachmen

Reviewer 5 Report
The aim of this paper is to describe the neuroprotective effects of Maresin1 in the septic rats focusing on the anti-inflammatory effects and microglial activation.
The topic of the manuscript is of current scientific interest. The experimental approach is also suitable. However, there are several points, especially within the material and methods section and results, that need to be addressed.
Introduction
Neuroendocrine what? neuroendocrine changes?
This sentence “the pathogenesis of sepsis is extensive and complex… ultimate organ diseases” describes sepsis in general it would be more fluent to insert it above and then describe the effects of sepsis in the brain.
To change “central inflammatory reaction”
To better describe the role of NF-kB and the link with the previously exposed data (cytokines and microglia); all this is not clear in the text.
To change the sentence of “its regulation on microglia” because the role of MaR1 has been analysed in the regulation on microglia in other in vivo and in vitro models which should be introduced in this section. Indeed, the effects of MaR1 in the brain are not described even if there are several publications about it.
Materials and methods
In the in vivo experiments approximate information has been made about the animal number used for different group and no information regarding the subdivision for the different experimental preparations.
Processing of tissues for immunohistochemistry and immunofluorescence analyses could be described without any repetition in the several paragraphs (2.5, 2.6 and 2.7). Dapi staining has been used not only in the tunnel. Quantitative analysis of staining should be appropriately described.
In situ detection of DNA fragmentation was performed using the TUNEL Detection System kit? which brand?
In the wb analyses all control groups have no statistics. how was the analysis done? is ß-actin used as standard to normalize? To describe in material and methods.
In the section 2.8 what is “Seahorse homogenate supernatant”? and the species of the primary antibodies should be specified.
In the section 2.5 and 2.8 describe the secondary antibodies used.
Results
In the section 3.1 the survival rate was not analysed in rats treated with MaR1 alone at different concentrations ... the results obtained should be indicated at least in the text.
Minor revision “Sham control” to replace with “sham group or control group”
In the section 3.3. The figure 3.A the image is low quality and low magnification. To indicate the layers of the hippocampal sections and show the same portions for all groups. Are the images representative? Arrows are not described anywhere.
To change the sentence “more active Iba-1-positive microglia “, the analysis suggests the increase of the number of Iba-1-positive microglial cells and not the activation state.
The figure 3D. the writings of the table are not visible
The legend of Figure 5 is not clear and there are diverse typing errors. Only in this western blot is used the ß-actin for quantitative analysis?
In the section 3.6 In the figure 6A the same picture is shown for the control and the treated group. This is a serious mistake. The staining image of Iba-1 alone in the LPS group shows positive immunofluorescence in the nucleus ...is it possible to give an explanation? To describe the scale bars and arrows. The treatment of MaR1 at lower concentration is not commented.
Was the analysis of wb done for Iba-1 in BV2 cells? This experiment might give strength to the conclusions
The Figure 6C, the image shown at this low magnification does not allow for this morphological analysis.

Round 2
Reviewer 3 Report
The authors revised the manuscript to my satisfaction
Reviewer 5 Report
The authors revised and improved their article with appropriate corrections.